# Ultrasensitive plasmonic sensing in air using optical fibre spectral combs

Christophe Caucheteur[1,*], Tuan Guo[2,*], Fu Liu[2], Bai-Ou Guan[2] & Jacques Albert[3]

Surface plasmon polaritons (SPP) can be excited on metal-coated optical fibres, enabling the accurate monitoring of refractive index changes. Configurations reported so far mainly operate in liquids but not in air because of a mismatch between permittivities of guided light modes and the surrounding medium. Here we demonstrate a plasmonic optical fibre platform that overcomes this limitation. The underpinning of our work is a grating architecture—a gold-coated highly tilted Bragg grating—that excites a spectral comb of narrowband-cladding modes with effective indices near 1.0 and below. Using conventional spectral interrogation, we measure shifts of the SPP-matched resonances in response to static atmospheric pressure changes. A dynamic experiment conducted using a laser lined-up with an SPP-matched resonance demonstrates the ability to detect an acoustic wave with a resolution of $10^{-8}$ refractive index unit (RIU). We believe that this configuration opens research directions for highly sensitive plasmonic sensing in gas.

[1] Department of Electromagnetism and Telecommunication, University of Mons, Boulevard Dolez 31, 7000 Mons, Belgium. [2] Guangdong Provincial Key Laboratory of Optical Fiber Sensing and Communications, Institute of Photonics Technology, Jinan University, 601 Huangpu Avenue West, Guangzhou 510632, China. [3] Department of Electronics, Carleton University, 1125 Colonel By Drive, Otrawa K1S 5B6, Canada. * These authors contributed equally to this work. Correspondence and requests for materials should be addressed to C.C. (christophe.caucheteur@umons.ac.be) or to T.G. (tuanguo@jnu.edu.cn) or to J.A. (Jacques.Albert@carleton.ca).

The use of surface plasmon resonance (SPR)[1,2] for gas detection and biosensing dates back to 1982 (refs 3,4). Since then, we have witnessed considerable research activities aimed at the development of optical sensors suited to measure (bio)chemical species. SPR-based sensors have demonstrated a great potential for affinity biosensors, yielding real-time analysis of specific bio-interactions, without using labelled molecules[5]. This technology has progressively emerged as the leading one in the field of direct and real-time monitoring of biomolecular interactions. The sensitivity to the surrounding refractive index often falls in the range of $10^{-6}$–$10^{-7}$ refractive index unit (RIU) for instruments with reference channels and temperature stabilization[6].

Currently, among the possible plasmonic optical sensor configurations, optical fibre-based SPR sensors present the highest degree of miniaturization. They have opened the path to remote monitoring by minimally invasive sensors that can be distributed or inserted into small spaces, otherwise unlikely to be reached with prism configurations. They have been extensively used in aqueous solutions.

The first fibre SPR configurations consisted of multimode optical fibres with their cladding locally removed or thinned to make the guided light interact with the metal coating deposited on the fibre surface[7]. SPR excitation efficiencies have been further enhanced with side-polished single-mode fibres[8]. In all cases, light-coupling is obtained either by etching or polishing, which requires a careful control and weakens the fibre. Hence, for practical reasons, the use of thick optical fibres (with cladding diameters between 200 and 400 µm) has been preferred[9].

Instead of removing the cladding, gratings photo-inscribed in the core of telecommunication-grade single-mode optical fibres can be used to diffract part of the light into the cladding. In this case, the resonant grating coupling only occurs at specific wavelengths, that is, different fibre modes couple distinctively at different wavelengths. This is similar to a coupled two-resonator system in which the first resonator consists of the grating that couples two fibre modes with each other and the second resonator is the metal coating that couples a fibre mode to a surface plasmon polariton (SPP). When these two resonances are matched, the wavelength of the corresponding grating resonances becomes sensitive to the changes in the SPR. Different grating architectures can be used. Uniform fibre Bragg gratings (FBGs) are narrowband, wavelength-selective filters that couple the forward-going core mode into a backward-going one. Light, therefore, remains confined in the fibre core and is insensitive to surrounding medium changes. Etched FBGs with the cladding removed have thus been used[10]. More advantageously, tilted FBGs (TFBGs) that have grating fringes slightly angled with respect to the normal of the optical fibre propagation axis couple light from the core towards the cladding while preserving the optical fibre integrity. Two kinds of coupling occur here: the self-backward coupling of the core mode (the Bragg resonance) and the backward coupling of the core mode with tens to hundreds of cladding modes. The TFBG-transmitted amplitude spectrum displays a dense comb of narrowband-cladding mode resonances (full-width at half-maximum ∼200 pm; ref. 11) for wavelengths at which light has been coupled out of the core. When a metal film is deposited on the cladding surface, the cladding modes that are phase-matched to the SPP tunnel energy into it and the spectrum is modified[12–14]. In the TFBG spectrum, the Bragg resonance is insensitive (in wavelength and power) to the surrounding refractive index changes. It can therefore be used to subtract unwanted temperature- and power-level fluctuation effects from the sensor response. Moreover, the grating tilt breaks the cylindrical symmetry of the fibre cross-section, allowing the separate excitation of high-order cladding modes that are radially polarized (transverse magnetic (TM) and EH modes) and azimuthally polarized (transverse electric (TE) and HE modes)[15]. Radially polarized mode resonances can excite TM-polarized SPP waves while the interleaved spectrum of azimuthally polarized ones cannot[16]. Fibre SPR refractometers with sensitivities of ∼500 nm RIU$^{-1}$ in aqueous solutions have been demonstrated[17]. Biochemical sensors based on the antibody/antigen affinity have also been reported for proteins and cell immunosensing[18–22]. They present limits of detection and sensitivities that compete with the most sophisticated plasmonic-based sensing solutions[23,24].

The development of optical fibre SPR platforms for gas sensing where the refractive index of the medium is far from the one of the fibre material is not trivial. Hence, unclad optical fibre and fibre-grating SPR configurations reported so far operate well in aqueous solutions but are not able to provide direct (that is, with only a single metal layer on the silica surface) SPP excitation in air or gaseous environments. Indeed, most of the configurations reported so far are unable to couple light into fibre modes that have sufficiently small effective refractive indices (corresponding to small enough incidence angles at the cladding outer boundary). Several configurations have been proposed to overcome this limitation. Multilayer coatings have been used on top of unclad fibre sections to provide a medium with the correct refractive index for SPP excitation[25–27]. Sensing mechanisms are based on the refractive index changes occuring in this multilayer structure when gases are adsorbed into it. A tapered fibre optic tip sensor with angled facets such that optical modes with effective index close to 1 can be excited has also been proposed[28,29].

In this paper, we demonstrate a robust all-fibre configuration able to excite SPP directly in air with narrowband-cladding mode resonances that can be measured with high resolution. It is based on highly tilted FBGs to allow the excitation of cladding modes with effective indices between 0.92 and 1.18 RIU, in gold-coated single-mode optical fibres. The SPP excited by this configuration is used in an experiment that reveals the potential of the technique in terms of absolute refractometric sensitivity in dilute gases. Here we rely on acoustically induced refractive index change sensing, as the latter requires an ultrahigh measurement resolution[30,31]. We demonstrate that the SPP-matched resonances can detect an acoustic wave moving across the fibre grating. This demonstration confirms that surrounding refractive index changes of $10^{-8}$ RIU can be resolved with a TFBG-SPR device.

## Results

**Transmitted amplitude spectrum of bare highly tilted FBGs.** The cladding mode resonances of a TFBG appear at well-defined wavelengths given by the following phase-matching conditions:

$$\lambda_{\mathrm{clad},i} = \left(n_{\mathrm{eff,core}} + n_{\mathrm{eff,clad},i}\right)\Lambda \qquad (1)$$

where $n_{\mathrm{eff,core}}$ denotes the effective refractive index of the core mode, $n_{\mathrm{eff,clad},i}$ the effective refractive index of the $i^{\mathrm{th}}$ cladding mode and $\Lambda$ the grating period measured along the optical fibre axis.

Figure 1 depicts the transmitted amplitude spectrum of a 16-mm-long 37° TFBG photo-inscribed in a single-mode optical fibre (Corning SMF-28) with a 1,073.81-nm-period uniform phase mask, yielding a Bragg resonance ∼1,550 nm (with an effective refractive index of 1.447). The internal tilt angle was 23°. Curves measured in different liquids are displayed on the same figure with a vertical offset. The cladding mode resonances on the short-wavelength side of the Bragg resonance can be divided into two main subsets:

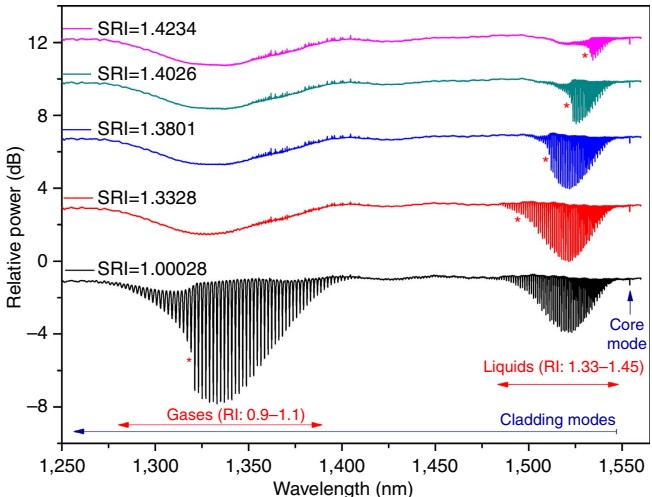

**Figure 1 | Transmitted power spectra of an uncoated 16-mm-long 37° TFBG in air and liquids.** Spectra measured for different surrounding refractive indices (SRIs) are plotted with an offset in the vertical scale. Red stars identify the cutoff wavelength in each medium.

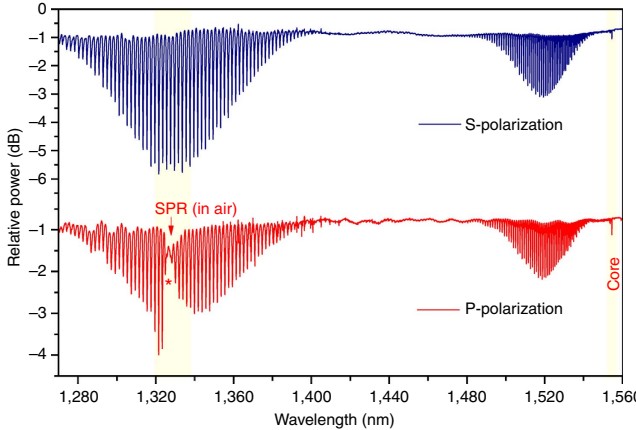

**Figure 2 | Evidence of SPR attenuation of cladding mode resonances in the P-polarized transmitted power spectra of a 37° TFBG coated with 50 nm of gold (red curve).** The resonances in the S-polarized spectrum (blue curve) at the same wavelengths near 1,325 nm are not attenuated. Shaded regions of the spectra emphasize the resonances of interest near the SPR (used for refractometric sensing) and near the Bragg resonance at 1,555 nm (for temperature compensation).

(1) in the wavelength range (1,480–1,550 nm), cladding mode resonances have effective refractive indices ranging between 1.30 and 1.44, and are therefore suited for measurement in aqueous solutions, similar to the case of weakly TFBGs with tilt angles limited to 10°.

(2) in the wavelength range (1,270–1,410 nm), cladding mode resonances present effective refractive indices ranging between 0.92 and 1.18, according to the aforementioned phase-matching condition.

The modes of this second subset, which can only be excited for tilt angle values above 30°, constitute the key elements of the proposed sensor configuration. Red stars in Fig. 1 indicate the position of the cutoff wavelength, corresponding to the cladding mode resonance for which its effective refractive index matches the one of the surrounding medium. Resonances at wavelengths shorter than the cutoff belong to leaky cladding modes.

**Behaviour of gold-coated highly-tilted FBG refractometers.** The transmission spectra of TFBGs, coated with a 50-nm-gold layer and left strain-free in air, are shown in Fig. 2. The input state of polarization was linear and oriented in the P-plane relative to the tilt direction to excite radially polarized light modes. A clear SPP signature (that is, the strong attenuation in the resonance amplitude because of the transfer and loss of power to the SPP) appears ~1,325 nm, slightly above the cutoff wavelength, according to Equation 1 and taking into account the optical fibre dispersion. This signature is comparable to the one already reported in aqueous solutions for weakly tilted FBGs. For S-polarized input light, the SPP cannot be excited by the azimuthally polarized cladding modes and no attenuation of the resonance amplitudes appears in the spectrum.

Validating simulation results can be plotted as the mode loss (imaginary part of the effective mode index) against the real part of the effective index, as shown in Fig. 3. There is a sharp increase in loss that is observed only for TM/EH modes in the vicinity of $n_{eff,clad(i)} = 1.007$, which is the value expected for the SPP wave-effective index of the gold/air boundary at these wavelengths. This loss increase leads to an attenuation of the resonance of the corresponding cladding mode. Simulations further indicate that the radial order of the modes with effective

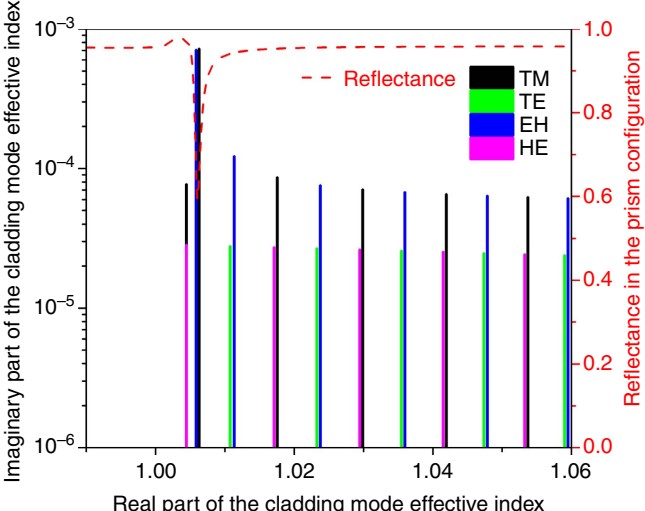

**Figure 3 | Simulation of the attenuation of gold-coated fibre-cladding modes with effective indices near 1.0.** Each vertical bar corresponds to a particular cladding mode and is colour-coded by its polarization state (TM and EH, which are P-polarized at the cladding surface; TE and HE, which are S-polarized). The dashed red line shows the theoretical reflectance of a conventional prism-based SPR device made from the same materials as the fibre (silica with a refractive index of 1.445), as a function of the effective refractive index of a TM surface wave propagating along the base of the prism during total internal reflection. A pair of P-polarized cladding modes with an effective index near 1.007 is phase-matched to an SPP and has an attenuation that is larger than that of its neighbours by one order of magnitude.

indices ~1.0 is near 167. Figure 3 also shows the theoretical reflectance of a gold-coated glass prism (with the same gold thickness and glass type as the fibre, that is, fused silica) as a function of the effective index of propagation of a surface wave propagating along the prism base. The latter curve has a reflectance minimum (because of energy transfer to a SPP of the gold interface) at the same value of effective index as the fibre

model. Finally, converting this effective index to resonance wavelength by Equation 1 and taking into account the dispersion of the glass index, the SPP resonance is predicted to occur near 1,325 nm, exactly where it is observed (Fig. 2).

Static gas-phase refractometry results are shown in Fig. 4a, which depicts the evolution of the SPP-matched cladding mode resonance when the atmospheric pressure is decreased in steps to reach ∼1,000 Pa. A strong blue shift of the mode resonance is obtained, accompanied by a change of its relative amplitude. Also shown in Fig. 4b is the spectrum in the vicinity of the Bragg resonance to show that it remains virtually unchanged, confirming that the evolution of the SPP mode arises from surrounding refractive index changes and not from physical effects (strain or temperature[32]).

Other measurements were performed on another grating with smaller atmospheric pressure changes to quantify the sensitivity of the SPP-matched cladding mode resonance to air pressure (and hence to the refractive index of the fibre surrounding). Figure 5 depicts a spectral region around the SPR signature, confirming that only the subset of modes that are phase-matched with the SPR envelope are sensitive to the atmospheric pressure change, with different degrees of sensitivity, as already observed for refractometic changes in liquids[33]. The inset of Fig. 5 focuses on the evolution of the most sensitive cladding mode resonance, showing again a clear wavelength shift and amplitude change. Similarly to our lines of work conducted in liquids, this mode is the most sensitive as it is located on the shoulder of the SPR envelope. Both the wavelength shift and amplitude change can be used to estimate the SPP modification induced by surrounding refractive index changes. The former measure is inherently insensitive to unwanted optical power fluctuations. The latter can also be made immune by self-referencing the spectrum with the Bragg resonance wavelength and power level, because they remain unaffected by surrounding refractive index changes, as demonstrated in Fig. 4.

The raw data of Fig. 5 were used to compute the sensitivity as a function of the surrounding refractive index change, knowing that the refractive index of air is linked to the atmospheric pressure and to the temperature as follows[34,35]:

$$n = 1 + (n_s - 1) \frac{p}{p_s} \frac{T_s}{T} \qquad (2)$$

where $n_s = 1.00026825$ (for dry air with 450 p.p.m. of $CO_2$), $p_s = 101{,}325$ Pa and $T_s = 288.15$ K.

Fig. 6a,b displays the corresponding evolutions of the wavelength shift and amplitude change as a function of the surrounding refractive index. A linear regression of the raw data yields a sensitivity of 204 nm RIU$^{-1}$ and 5,515 dB RIU$^{-1}$,

respectively. These values agree well with those reported for gold-coated weakly TFBG refractometers used in aqueous solutions.

**High-resolution refractometric sensing with highly-tilted FBGs.** Finally, dynamic measurements have been performed with the experimental setup described in the Methods section. The loudspeaker and the sensor were placed on top of two different vibration-isolated tables to make sure that the sensor measures only the atmospheric pressure change induced by the sound waves in air. The distance between the loudspeaker (6 cm diameter without acoustic focusing) and the sensor was 15 cm. The amplifier was set close to its maximum so that the sound level $L$ was measured to be 109 dB at the sensor location. The sound level $L$ is defined by the following relationship:

$$L = 20 \log \frac{p}{2e^{-5}} \qquad (3)$$

with $p$ expressed in Pa (ref. 36).

According to the aforementioned formula, a sound level of 109 dB yields an atmospheric pressure change of only 5.6 Pa, corresponding to an air refractive index change as small as $1.48 \times 10^{-8}$ RIU at constant temperature (Equation 2).

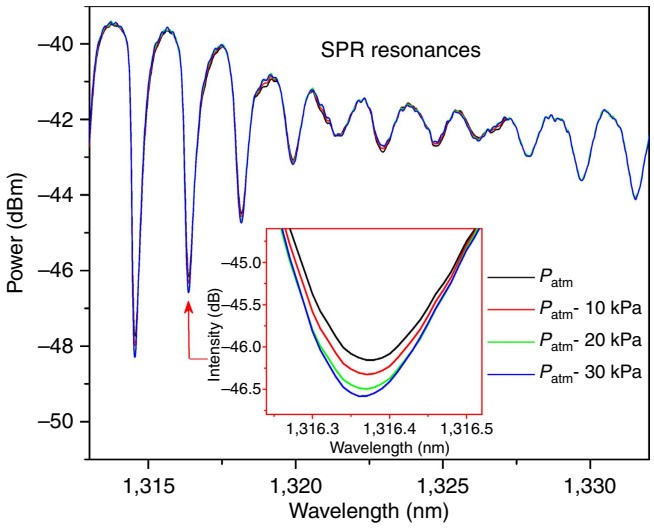

**Figure 5 | P-polarized transmitted power spectrum evolution due to atmospheric pressure changes.** The inset shows the evolution of the most sensitive cladding mode resonance (centred at 1,316.37 nm) among those within the SPP attenuation band (∼1,312–1,332 nm).

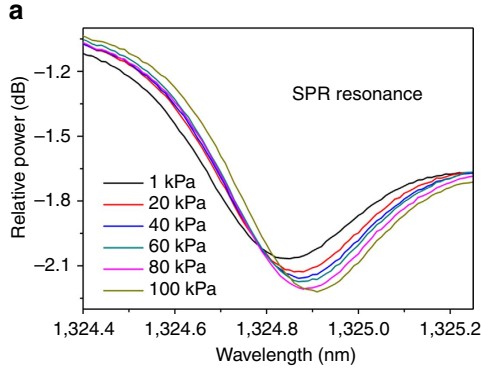

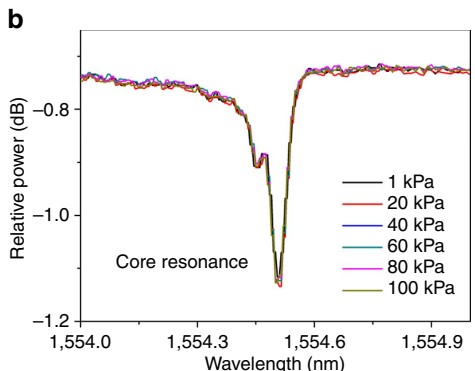

**Figure 4 | Effect of atmospheric pressure changes on the resonances of interest of a gold-coated highly tilted FBG.** (**a**) The evolution of an SPP-matched cladding mode resonance in response to strong changes of the atmospheric pressure. (**b**) Focuses on the core-mode resonance.

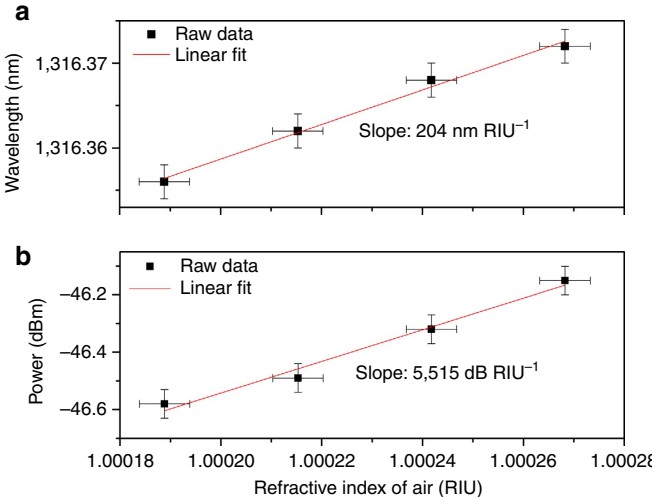

**Figure 6 | Sensor response to a change of the refractive index of air.** Evolution of the wavelength (**a**) and minimum power level (**b**) of the most sensitive cladding mode resonance as a function of the SRI variation due to atmospheric pressure changes. Error bars result from the uncertainties in the determination of the cladding mode resonance ($\pm 2$ pm), power level ($\pm 0.05$ dBm) and atmospheric pressure ($\pm 1$ kPa).

The tunable laser wavelength was then set to correspond to different cladding mode resonances of interest to assess their behaviour in response to such a small surrounding refractive index change. The laser has a linewidth of 500 kHz (or $\sim 3$ fm at 1,320 nm) and its wavelength was positioned on the edge of a given cladding mode resonance, in order for resonance shifts to cause the largest possible transmission-level changes. Figure 7 displays the typical experimental responses for several resonances, obtained for a 2-kHz sine excitation. It shows that the most sensitive cladding mode resonance (located at a wavelength of 1,316.382 nm, with the probe laser tuned to 1,316.270 nm) follows the sine wave excitation accurately and therefore that it yields an easily measurable response to refractive index changes of the order of $10^{-8}$. The oscillation of the neighbouring mode (at 1,318.184 nm) is three times less sensitive but is still able to follow the sound wave. Other modes in the same spectral region were not sensitive enough and are therefore not displayed in this figure. The other mode traces in Fig. 7 correspond to the azimuthally polarized mode closest to the SPP (and located at 1,317.024 nm) and to the ghost-mode resonance (1,542.288 nm) that is adjacent to the Bragg resonance. The ghost-mode resonance is the most sensitive resonance to bending[11], and the fact that these two resonances remain clearly in the noise level of the photodiode (PD) confirms that the reported signals are only due to the perturbation of the SPP-effective index by air refractive index variations from the acoustic wave. The key point of this ultralow detection level is that the measurement is dynamic (that is, not disturbed by lower-frequency drift) and especially that the TFBG resonances are very narrow and deep, and therefore have large slopes where a stable ultranarrow linewidth laser can be used to probe very small shifts of selected cladding mode resonances.

To determine the limit of detection of the reported sensing configuration, another set of experiments was conducted by tuning the sound level by steps of $\sim 1$ dB between 105 and 109.2 dB for a 2-kHz sine excitation. The latter value is the maximum sound level that our amplified source can reach. For a sound level below 105 dB (this value corresponds to an atmospheric pressure change of 3.56 Pa and an air refractive index change of $0.94 \times 10^{-8}$ RIU), the sensor trace recorded by the oscilloscope remained flat. A growing sine function at

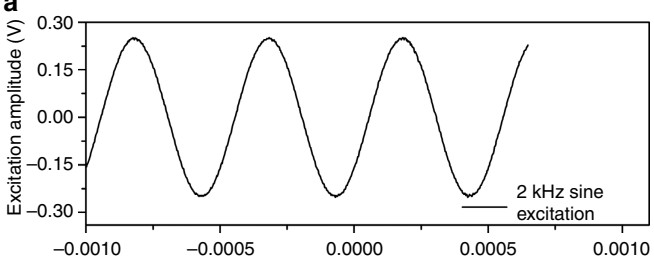

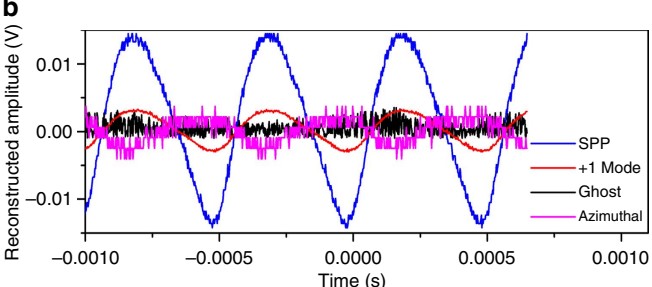

**Figure 7 | Sensor response to an acoustic wave.** (**a**) The waveform of the 2 kHz sine wave excitation source for the sound wave. (**b**) The photodetector measurement of the transmitted power when a single-wavelength tunable laser source is lined-up on the edge of several selected resonances: SPP phase-matched mode at 1,316.382 nm (blue curve), nearest neighbour mode at 1,318.184 nm (red curve), ghost mode at 1,542.288 nm (black curve) and S-polarized mode closest to the SPP at 1,317.024 nm (magenta curve).

2 kHz frequency was then recorded for sound levels above 106 dB, as depicted in Fig. 8a. For each investigated sound level, five measurements were recorded, allowing us to compute the mean value and s.d. of the peak-to-peak amplitude of the recorded sine evolution. Figure 8b depicts the obtained results as a function of the surrounding refractive index change, which was computed from the measurement of the sound level using Equations 2 and 3. From these results, we can confirm that the limit of detection of our sensor configuration is very close to $10^{-8}$ RIU.

## Discussion

The transmission spectrum of the TFBG in air shown in Fig. 1 indicates that cladding modes with effective indices lower than 1.0 are excited, and therefore that the evanescent surface wave has a superluminal velocity in the axial direction of the fibre. In more detail, the effective refractive index of a mode is given by the ratio of the phase velocity in vacuum over the phase velocity of the mode in the fibre, measured along the fibre axis. Taking the analogy of a plane wave incident on a flat interface between a high refractive index medium and a low refractive index medium, a similar effective index can be calculated for the phase velocity along the interface. As the angle of incidence decreases from 90 degrees towards the critical angle, the phase velocity along the interface increases (and the effective index decreases). Passing the critical angle, the wave becomes leaky but the phase velocity along the interface continues to increase. The same phenomenon occurs in fibres, or at the base of a Kretschmann–Raether prism, as the angle of incidence decreases (equivalent to the order of the cladding mode increasing). For a fibre in air (or vacuum), effective indices lower than 1.0 mean that the corresponding cladding mode is leaky (but still propagating).

Temperature changes were not considered in this work for which dynamic measurements were conducted in a thermally

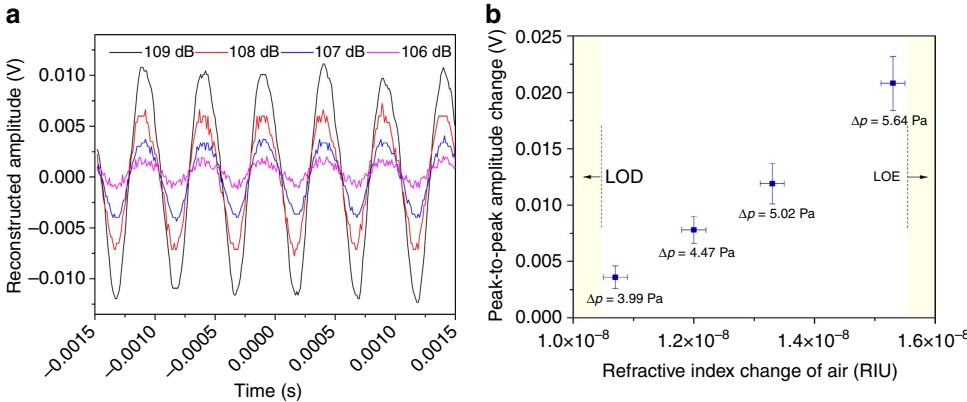

**Figure 8 | Determination of the limit of detection (LOD) of the sensor.** (**a**) The time response of the photodetector for four different sound levels (colour-coded in dB) with the probe laser tuned to 1,316.270 nm, on the edge of the SPP resonance centred at 1,316.382 nm. (**b**) The peak-to-peak photodetector voltage change as a function of the surrounding refractive index modulation amplitude (calculated from the air pressure modulation of each sound level). Vertical error bars show the s.d.'s from five independent measurements. Horizontal error bars represent the error made on the computation of the refractive index value due to the uncertainty on the sound measurement (0.1 dB). The term LOE means limit of excitation (that is, the maximum sound level available with our system).

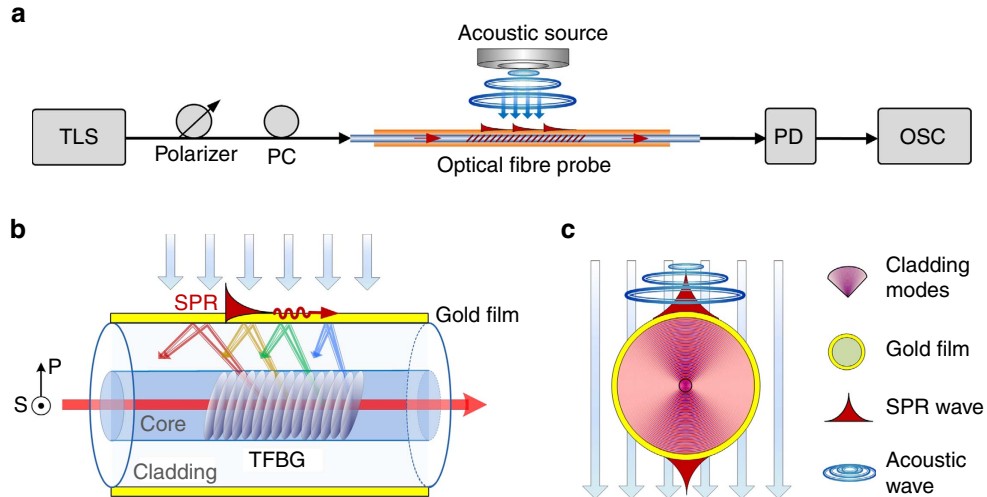

**Figure 9 | Acoustic wave-sensing principle with gold-coated highly tilted FBGs.** (**a**) The experimental setup used to interrogate the sensor. OSC, oscilloscope; PC, polarization controller; PD, photodiode; TLS, tunable laser source. (**b**) SPR excitation mechanism with gold-coated highly tilted FBGs. S and P indicate the polarization direction relative to the fibre. (**c**) Acoustic wave influence of the SPR wave.

stabilized clean room environment over limited periods of time not exceeding a few seconds. In such small time periods, the ambient temperature can be considered as constant. Note that the influence of the temperature on the fibre itself (because of the thermo-optic coefficient of glass and the thermal expansion of the grating period) is removed from the measurements by referencing all wavelengths to the resonance of the core-mode reflection (that senses temperature and strain but not SRI)[11]. This can be carried out by locking the wavelength of the probe laser at a precise offset from Bragg wavelength with a suitably fast feedback loop system.

The observations reported in this paper constitute an experimental evidence that SPP waves can be excited on the surface of a metal-coated fibre in air at atmospheric and lower pressures. The device uses a TFBG with a tilt angle near 37° to excite cladding modes that have effective indices from 0.92 to 1.18 RIU, and a polarization state that is oriented to maximize the coupling of energy from the cladding to the SPP (that is, radially at the cladding boundary). It was further demonstrated

that such a device can detect changes in the refractive index of air as the pressure is varied, with sensitivities of 204 nm RIU$^{-1}$ and 5,515 dB RIU$^{-1}$. Most importantly, the use of a single-wavelength, narrow linewidth interrogation technique was shown to enable this device to measure refractive index changes of the order of $10^{-8}$ RIU dynamically at 2 kHz. This achievement stems from many factors, but mostly from the fact that the power transmission spectrum of TFBGs contains a low insertion loss, very dense frequency comb of narrowband resonances that can 'probe' many kinds of perturbations to the fibre and its surroundings. Because of the differential selectivity of the modes to different types of perturbations, most of the spectral comb typically remains invariant under a specific perturbation and can be used as a power level and wavelength reference to reduce system measurement noise. Finally, the main enabling advance reported here was the successful extension of this frequency comb to cover fibre modes with effective indices above and below 1.0, thus allowing for SPP excitation in gases at any pressure down to vacuum levels.

As a consequence, the proposed configuration has the potential to be used for highly sensitive gas detection (at p.p.m. levels or even lower), with specific receptors grafted on the metal surface. We therefore believe that it opens up prospects for the development of remotely operated plasmonic optical fibre sensors in gaseous environments and also, more generally, as a means to generate plasmon waves on optical fibre surfaces with various materials including graphene[37,38].

## Methods

**Fabrication of gold-coated highly tilted FBGs.** Highly-tilted FBGs ($\sim$15–20 mm in length) were manufactured using the phase-mask technique in photosensitive single-mode fibres, similarly to[39,40]. The fabrication process mainly includes the following three steps. First, the fibre was hydrogen-loaded (temperature: 50 °C, pressure: 1,500 p.s.i., loading time: 168 h) to increase the photosensitivity of the fibre core. Second, pulsed 193 nm ultraviolet light (power of 3 mJ per pulse, frequency of 200 Hz) was focused with a cylindrical lens and scanned over the fibre in the axial direction. Finally, a large tilt angle (over 35 degree) was introduced by rotating the phase mask and fibre consistently around an axis perpendicular to the laser beam (with the phase mask and fibre kept parallel). The actual tilt angle of the grating planes inside the fibre is decreased by refraction at the air–glass interface. It should be noted that, with the tilt angle increase, the grating inscription efficiency is decreased. It was empirically found that precisely positioning the fibre within the ultraviolet focusing plane, slowing down the scanning speed and using a cylinder lens with a long focus length are the key elements to achieve strong and highly-tilted FBGs.

A 50-nm-thick gold film was deposited on the fibre probe by radio frequency (RF) magnetron sputtering. To achieve a high-quality coating, two issues must be noted. One is to improve the film adhesion by a 2–3-nm-thick chromium layer sandwiched between the optical fibre surface and the gold film. The second is to ensure the uniformity of the gold film thickness around the fibre circumference by rotating the fibre device along its axis during deposition.

**Fibre-mode simulations.** Grating spectra can be studied and predicted using coupled mode theory[41]. In our case, numerical simulations conducted using the finite difference mode solver FimmWave (from Photon Design Inc.) were carried out to confirm the experimental evolutions. FimmWave calculates the complex effective refractive indices of all the modes supported by the fibre geometry used in this work.

**Static amplitude spectrum measurements.** Transmitted power spectrum measurements were obtained with an unpolarized super wideband light source (Amonics ASLD-CWDM-5-B-FA) and an optical spectrum analyser (Yokogawa AQ6370). An in-line polarization controller from General Photonics was placed upstream of the TFBG to control the input state of polarization. The correct polarization state is identified by observation of the spectrum during polarization adjustment[11,13,14]. Note that there is not only one polarization state that yields SPP excitation but rather a set. For instance, when linear polarization states are used, a range of 10° can be used for clean SPP excitation in the amplitude-transmitted spectrum of the TFBG.

Static atmospheric pressure measurements were conducted in a custom-made enclosure in which controlled pressure changes can be applied and measured using a graduated water column. A 2-mm-thick rubber seal containing two small apertures (250 μm) was used to avoid air leaks while ensuring that the connecting fibres used for the real-time monitoring are not subjected to transverse forces when the pressure is modified in the chamber.

**Dynamic acoustic wave sensing.** Acoustic wave sensing was obtained with a dynamic interrogation setup composed of a tunable laser source (TLS, Santec TLS-210V) followed by the aforementioned polarization controller and a PD (New Focus Model 2011). The laser wavelength was matched to a SPP-active mode resonance by simultaneous measurement with the optical spectrum analyser. The electrical output of the PD was connected to an oscilloscope (Agilent DS06012A (100 Mz)) to record the time evolution of the output signal. Acoustic waves were produced by a loudspeaker connected to an audio amplifier (QSC RMX850) that was driven by a 20-MHz function generator (Agilent 33220A). Sinusoidal functions with a frequency ranging between 1 Hz and 3 kHz were used in our experiments. The corresponding sound level in dB was measured using an XL2 analyser from NTI Audio. This experimental setup is sketched in Fig. 9a. The bottom part of Fig. 9 shows the operating principle of SPR generation in gold-coated TFBGs (Fig. 9b) and the preferred optical fibre orientation with respect to the acoustic wave (Fig. 9c).

**Data availability.** The data that support the findings of this study are available from the corresponding authors upon request.

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

## Acknowledgements

This work was supported by the Belgian F.R.S.-FNRS (Associate research grant of C.C.), the European Research Council (Starting grant of C.C.—Grant agreement No. 280161), the ARC (Actions de la Recherche Concertée de la Fédération Wallonie-Bruxelles) Prediction project, the Guangdong Youth Science and Technology Innovation Talents of China (No. 2014TQ01X539), the Guangdong Natural Science Foundation of China (No. 2014A030313387), the Guangdong Innovation Foundation of China (No. 2015KTSCX016), the Guangzhou Key Collaborative Innovation Foundation of China (No. 201504290942056) and by the Natural Sciences and Engineering Research Council of Canada RGPIN 2014-05612.

## Author contributions

F.L. and T.G. conceived the sensors. C.C. and T.G. conceived and carried out the experiments and analysed the data. B.-O.G. and J.A. supervised the project and analysed the data. Theoretical analyses were carried out by C.C. and J.A. All authors contributed to the preparation of the manuscript.

## Additional information

**Competing financial interests**: The authors declare no competing financial interests.

**How to cite this article**: Caucheteur, C. *et al.* Ultrasensitive plasmonic sensing in air using optical fibre spectral combs. *Nat. Commun.* **7,** 13371 doi: 10.1038/ncomms13371 (2016).

**Publisher's note**: 

