## [Peer review file · Nature Communications]

Reviewers' comments:

Reviewer #1 (Remarks to the Author):

The work here proposed for publication brings some new inputs and it is a highlight in the SPR wide area of research. The high potential of SPR-based fiber optic sensors, for the achievement of high resolutions/sensitivities, has drawn a strong line of research regarding its application in liquids and gas sensing.

1. In fact, the achievement of an unprecedented sensitivity of 10^{-8} RIU with the proposed SPR-based sensor configuration for gas sensing is the major claim of this work. However, this result raises some questions that should be discussed:

a) The authors use the term "sensitivity" inaccurately (also used in the state of the art). It's resolution instead.

b) Although being a high resolution, it's not clear at all how this value was obtained (numerically, I presume). The great benefit of this work would be if such resolution had been obtained experimentally (for example, by a simple experiment based on a RI step-change).

c) The state of the art only mentions works where RI was detected in liquid media; however there are many reports on the measurement of RI or pressure in gaseous media (ex: S. K. Mishra, S. Bhardwaj, and B. D. Gupta, "Surface Plasmon Resonance-Based Fiber Optic Sensor for the Detection of Low Concentrations of Ammonia Gas" IEEE Sensors Journal, vol. 15, no. 2, February 2015).

d) This raises another question which is the parameter measured in the experiment. When we have a gaseous medium it is expectable to obtain a resolution in the order of ppm or ppb; resolutions in RIU units are usually seen for liquid media. So, what is the great importance of the obtained resolution in terms of plasmonic use for gas sensing?

2. Another important issue that was not considered in the experimental setup is the referencing of the optical power transmitted by the broadband source. This should be justified as well.

3. The last sentence of "introduction" requires a reference: "such an assessment cannot be made in liquids, (...) temperature constant to 10^{-4} degrees".

4. Typing errors: figures 5 and 6: cladding instead of caldding.

In my opinion, this work does not fulfill the requirements to be published in Nature Communications.

Reviewer #2 (Remarks to the Author):

General comments:

- The authors assert that they have developed a SPP sensor that operates in air, and is therefore capable of detecting changes to refractive indices close to unity. This is interesting; however there are existing publications that report fiber plasmonic devices that operate in air, such as T. Allsop, et. Al., "Exploitation of multilayer coatings for infrared surface plasmon resonance fiber sensors," Appl. Opt. 48, 276-286 (2009). Recent work from that author also shows research for the detection of gasses with indices marginally above unity.
- I would argue that the sensor architecture is not in itself new, as there are numerous references on the use of tilted gratings in metal coated fibres, as has been noted by the authors in their reference listings. How much tilt constitutes an original architecture? I would argue that tilt alone does not. However, the act of tilting does extract unique sensing properties that the authors have exploited.

- In general, the referencing is too self-focused, for example, there are numerous journal and conference papers on highly tilted fiber Bragg gratings (tilt $> 45^\circ$) that have not been referenced, admittedly without necessarily being for the purpose of plasmon generation.
- The authors state that refractive indices less than unity only occur for specific conditions, do the authors meet these conditions? Yes I believe that they do show this to be the case.
- Measurement of acoustic wave with 10-8 RIU sensitivity is interesting, but I am troubled about the claimed resolution and a missing graph that I expected.

Page 3, para 3 and 4:

I would suggest that reference 8 is not applicable to the introductory text discussing D-shaped fibers and plasmons, but rather M. Zervas and I. Giles, "Performance of surface-plasma-wave fiber-optic polarizers," *Opt. Lett.* 15, 513-515 (1990), is a more suitable reference.

I would also dispute that polishing weakens fibers, as there are many examples of quality polished components where fibers are embedded in glass blocks, prior to polishing, forming a robust, monolithic structure after polishing. Does the fiber, which the authors use, not have to be stripped to be coated with Au, where is the robust nature here? I think that general statements of this type in support of an apparent advantage rarely offer anything.

I think that the use of the expression "privileged" when discussing reference 9, it is not used conventionally and is confusing.

Page 4

Last paragraph is incorrect, there are fibre geometries that operate in air and other gasses, see general comments and references noted above regarding this.

Page 5

OK

Page 6

General information is OK here and of sufficient detail, although a comment regarding how accurately the polarization "axes" must be aligned to would be necessary; again I think there are other papers to reference in addition to [11]. How is the system affected by slow/sudden polarization changes? How stable is the sensor over time?

That word "privileged" appears again.

Figure 1: Good figure.

Page 7

Nice explanation and crystal clear.

Figure 2: Good figure.

Page 8

Good explanation, good figure (Fig. 3).

Page 9

Figure 5, "cladding" spelling error. I would also suggest that the comparative figures should have the same scale on the wavelength axes.

Page 10

Figure 6, "cladding" spelling error.

In Fig 5 the authors show the activated cladding mode, for which they also perform numerical analysis as a form of "confirmation", is located at 1325nm, but in the measurements of Fig 6 they select a cladding mode close to 1316nm. It is critical to define which modes are selected and why, is a wavelength shift preferential or a loss measurement? Surely the former mitigates any intensity fluctuations in the system, which could prove very important, as the changes in amplitude (and wavelength) are small. If the intensity change is self-referenced then that should be made clear. Whatever the case an explanation would be useful.

The final experiment is crucial to this work; it is the justification for this paper. However, here I am puzzled by the results that are presented. Let me explain why. Fig 6 shows the most sensitive response to pressure for which a 30kPa pressure change results in an intensity change of < 1dB, it looks closer to 0.5dB, or 1.67×10^{-5} dB/Pa, or $\sim 10^{-4}$ RIU. The raw data is plotted in Fig 7 and the error bars suggest an error far greater than the claimed measurement resolution. The source of the errors is critical but not stated, although it could possibly be linked to the ability to control the pressure and/or temperature during the experiment. Hence this should be addressed.

Now an experiment is undertaken where the pressure change is limited to 5.6Pa or an equivalent intensity change of 9×10^{-5} dB, or $\sim 1.8 \times 10^{-8}$ RIU. This may be anywhere from 100 to 1000 times smaller than the error estimation of Fig 7. Another high-resolution measurement method is used to record the low pressure data. The data in Fig 8 looks good with the two curves tracking each other, but where is the graph showing amplitude (pressure change) versus RIU? I would suggest that such a graph is paramount for the low pressure scale (that would also have the "same" gradient as Fig 7 if all is correct and calibrated) and would be most appropriate in order to make this final, and most important, result convincing.

Furthermore, more information is required regarding the speaker location, acoustic focussing and positioning of the sensor relative to the acoustic wave.

To summarise, the uniqueness of this work relies on the use of SPR cladding modes generated by highly tilted gratings, to measure low indices and hence pressure. The work draws extensively on earlier research by the same authors to develop their sensor system. The key result is an acoustic measurement for a low pressure wave that is associated with a RIU resolution of $\sim 10^{-8}$. I believe that some more work is required to remove any possible sources of error in this very delicate and potentially tricky experiment. This concludes my review.

Reviewer #3 (Remarks to the Author):

The authors C. Caucheteur et al propose and demonstrate highly sensitive surface plasmon resonance (SPR) sensors in air by using standard single mode optical fibers inscribed with tilted Bragg gratings. Based on the tilted Bragg gratings, SPR can be excited at the interface between the optical fiber cladding and the metal coating, which is extremely sensitive to the refractive index (RI) changing at external RI of around 1.0. Attributing to the narrow transmission spectrum of tilted Bragg gratings, the sensitivity of the proposed SPR-based sensor to external gas RI can be significantly improved, which is about 10^{-8} RIU near the RI of 1.0. To the best of our knowledge, such sensitivity is the highest one based on SPR technique.

In the background introduction, the authors emphasize that the fiber-based SPR sensors have never used in gaseous phases (line 4, paragraph 3, page 3). However, some structures like prism-coupling have been introduced into optical fiber for the vapor phase sensors, as shown in the following references. But the reported sensitivity is relatively low due to the wide band SPR spectrum.

Ref.1 Yoon-Chang Kim, Soame Banerji, Jean-Francois Masson, Wei Peng and Karl S. Booksh, Fiber-optic surface plasmon resonance for vapor phase analyses, *Analyst*, 2005, 130, 838-843.

Ref.2 Yoon-Chang Kim, Wei Peng, Soame Banerji, and Karl S. Booksh, Tapered fiber optic surface plasmon resonance sensor for analyses of vapor and liquid phases, *OPTICS LETTERS*, 2005, 30(17):2218-2220

The explanation of two-resonator system is unclear. The authors should give which order cladding

modes are excited in the SPR wavelength band. Because the effective RI of any fiber modes cannot be less than 1, the detailed explanation about the cladding mode resonance is of importance to convince the reader that the cladding mode resonances can present effective RI ranging between 0.92 and 1.18.

The temperature as well as the polarization perturbation to the accuracy of sensors' sensitivity should be clearly illustrated. As given in Eq. (2) in page 10, the RI of air is inversely proportional to the ambient temperature (T). Does the crosstalk from temperature influence the SPR cladding mode resonance? In addition, the proposed titled FBG is polarization-sensitive. Does the polarization perturbation affect the SPR cladding mode resonance and thus the sensors' sensitivity?

Further discussion about the limitation of the used equipment on the accuracy of sensors' sensitivity is required. The wavelength shift of the proposed SPR sensor is less than 0.01 nm as shown in Fig. 7, however, the highest resolution of optical spectrum analyzer (AQ6370) is 0.01 nm. Do the wavelength resolution limitation of OSA and the sensitivity of power-meter influence the accuracy of the obtained sensors' sensitivity?

The typos need to be revised:

"1.48 10⁻⁸" in line 7, paragraph 1, page 11

"caldding" in the annotation of Fig. 5 and Fig. 6

Reviewer #1 (Remarks to the Author):

The work here proposed for publication brings some new inputs and it is a highlight in the SPR wide area of research. The high potential of SPR-based fiber optic sensors, for the achievement of high resolutions/sensitivities, has drawn a strong line of research regarding its application in liquids and gas sensing.

1. In fact, the achievement of an unprecedented sensitivity of 10^{-8} RIU with the proposed SPR-based sensor configuration for gas sensing is the major claim of this work. However, this result raises some questions that should be discussed:

a) The authors use the term "sensitivity" inaccurately (also used in the state of the art). It's resolution instead.

Agreed. The term *sensitivity* defines the sensor response to the measurand, usually expressed in nm/RIU for SPR wavelength shift. Here, we have demonstrated that the sensor can resolve surrounding refractive index changes as small as 10^{-8} RIU and the correct term is therefore *resolution*.

b) Although being a high resolution, it's not clear at all how this value was obtained (numerically, I presume). The great benefit of this work would be if such resolution had been obtained experimentally (for example, by a simple experiment based on a RI step-change).

We are sorry that we were not clear enough. The high resolution was demonstrated during an experiment for which the gold-coated highly tilted fiber Bragg grating was exposed to a sound (sine wave) emitted by a loudspeaker. As the sound level was measured experimentally, the corresponding refractive index change was computed using well-known formulas. For a sound level of 109 dB, this yields a refractive index change of $1.48 \cdot 10^{-8}$ RIU. The latter can be detected by the SPP mode, as the blue curve of Fig. 8 clearly follows the sine excitation.

To clarify this point, we have modified Page 11 as follows: *The amplifier was set close to its maximum so that the sound level (corresponding to $20 \log \frac{p}{2e^{-5}}$ with p expressed in Pa [25]) was measured to be 109 dB at the sensor location, using the XL2 analyzer from NTI Audio. According to the aforementioned formula, this yields an atmospheric pressure change of only 5.6 Pa, corresponding to an air refractive index change as small as $1.48 \cdot 10^{-8}$ RIU at constant temperature (Eq. 2).*

c) The state of the art only mentions works where RI was detected in liquid media; however there are many reports on the measurement of RI or pressure in gaseous media (ex: S. K. Mishra, S. Bhardwaj, and B. D. Gupta, "Surface Plasmon Resonance-Based Fiber Optic Sensor for the Detection of Low Concentrations of Ammonia Gas" IEEE Sensors Journal, vol. 15, no. 2, February 2015).

Agreed. This comment is also shared by the other two reviewers. The introduction has been modified as follows: *The development of optical fiber SPR platforms for gas sensing where the refractive index of the medium is far from the one of the waveguide is not trivial. Hence, unclad optical fiber and fiber grating SPR configurations reported so far operate well in aqueous solutions (i.e. in any media with refractive indices between 1.30 and ≈ 1.43) but are not able to provide direct (i.e. with only a single metal layer on the silica surface) SPP excitation in air or gaseous environments. Indeed, most of the configurations reported so far*

are unable to couple light into fiber modes that have sufficiently small effective refractive indices (corresponding to large enough incidence angles at the cladding outer boundary). Several configurations types have been proposed to overcome this limitation. Multilayer coatings have been used on top of unclad fiber sections to provide a medium with correct refractive index for SPP excitation [24-26]. Sensing mechanisms are then based on the refractive index changes that occur in this multilayer structure when gases are adsorbed into it (the adsorption can be chemically specific depending on the material that is used), which in turn will modify the SPP. A tapered fiber optic tip sensor with angled facets such that optical modes with effective index close to 1 can be excited has also been proposed few years ago [27,28].

In this paper, we demonstrate a quite simple and robust all-fiber configuration able to excite SPP directly in air with narrowband cladding mode resonances (linewidth < 200 pm) that can be measured with a high resolution.

d) This raises another question which is the parameter measured in the experiment. When we have a gaseous medium it is expectable to obtain a resolution in the order of ppm or ppb; resolutions in RIU units are usually seen for liquid media. So, what is the great importance of the obtained resolution in terms of plasmonic use for gas sensing?

Our work is the first that shows a direct measurement of the refractive index of a gas by SPR on a metal coated fiber without the use of an intermediate coating. One of the uses, which is actually demonstrated in our manuscript, is the detection of sound with a device that has no moving parts. Furthermore, unlike the device recently highlighted in Nature Photonics (page 356, NATURE PHOTONICS | VOL 10 | JUNE 2016, our system does not rely on interferometry to achieve high sensitivity.

2. Another important issue that was not considered in the experimental setup is the referencing of the optical power transmitted by the broadband source. This should be justified as well.

Good point. When a broadband optical source is used, the measurement principle is based on the recording of the transmitted amplitude spectrum. In this case, the referencing can be easily made using the baseline level around the Bragg wavelength, which is not sensitive to surrounding refractive index medium changes. Referencing is more critical when using the narrowband laser source where in this case intensity measurements are obtained using a photodiode. In an optimum set-up, the referencing can be provided with two photodiodes (or a photodiode with two inputs) with one measuring directly the optical power emitted by the source, so that its power fluctuations over time can be subtracted from the sensor response. In our case, the measurements were dynamic and carried out during a limited time period of maximum 1 second. Referencing is therefore not an issue in our case.

3. The last sentence of "introduction" requires a reference: "such an assessment cannot be made in liquids, (...) temperature constant to 10^{-4} degrees".

This information stems from years of experimentation on this topic in our laboratories. It would be cumbersome and lengthy to address this point fully. As it is not essential to the discussion of the current results, the sentence in question has been removed from the introduction.

4. Typing errors: figures 5 and 6: cladding instead of caldding.

These typos have been corrected.

In my opinion, this work does not fulfill the requirements to be published in Nature Communications.

Reviewer #2 (Remarks to the Author):

General comments:

- The authors assert that they have developed a SPP sensor that operates in air, and is therefore capable of detecting changes to refractive indices close to unity. This is interesting; however there are existing publications that report fiber plasmonic devices that operate in air, such as T. Allsop, et. Al., "Exploitation of multilayer coatings for infrared surface plasmon resonance fiber sensors," Appl. Opt. 48, 276-286 (2009). Recent work from that author also shows research for the detection of gasses with indices marginally above unity.

Agreed. The introduction has been modified to figure out this: *The development of optical fiber SPR platforms for gas sensing where the refractive index of the medium is far from the one of the waveguide is not trivial. Hence, unclad optical fiber and fiber grating SPR configurations reported so far operate well in aqueous solutions (i.e. in any media with refractive indices between 1.30 and ≈ 1.43) but are not able to provide direct (i.e. with only a single metal layer on the silica surface) SPP excitation in air or gaseous environments. Indeed, most of the configurations reported so far are unable to couple light into fiber modes that have sufficiently small effective refractive indices (corresponding to large enough incidence angles at the cladding outer boundary). Several configurations types have been proposed to overcome this limitation. Multilayer coatings have been used on top of unclad fiber sections to provide a medium with correct refractive index for SPP excitation [24-26]. Sensing mechanisms are then based on the refractive index changes that occur in this multilayer structure when gases are adsorbed into it (the adsorption can be chemically specific depending on the material that is used), which in turn will modify the SPP. A tapered fiber optic tip sensor with angled facets such that optical modes with effective index close to 1 can be excited has also been proposed few years ago [27,28].*

- I would argue that the sensor architecture is not in itself new, as there are numerous references on the use of tilted gratings in metal coated fibres, as has been noted by the authors in their reference listings. How much tilt constitutes an original architecture? I would argue that tilt alone does not. However, the act of tilting does extract unique sensing properties that the authors have exploited.

It is not so much the amount of tilt that is highly novel here, but rather the fact that for the tilt angle we used, cladding modes with effective indices close to 1.0 (above and below this value in fact) are excited efficiently and provide narrowband resonances in the spectrum. Indeed, the key point behind our work is to use narrowband cladding mode resonances that can probe the surrounding medium with a high resolution, since narrowband resonances can be measured much more accurately than broad ones (as in the case of unclad optical fibers). To obtain resonances with an effective refractive index matching the one of air or gaseous media, high tilt angles are necessary.

- In general, the referencing is too self-focused, for example, there are numerous journal and conference papers on highly tilted fiber Bragg gratings (tilt $\gg 45^\circ$) that have not been referenced, admittedly without necessarily being for the purpose of plasmon generation.

This has been improved through the modification of the introduction (see answer to first comment above) and by adding two references in the beginning of the Methods section (refs 30 and 31).

- The authors state that refractive indices less than unity only occur for specific conditions, do the authors meet these conditions? Yes I believe that they do show this to be the case.

As illustrated by Figures 3 and 4, we are indeed meeting these conditions.

- Measurement of acoustic wave with 10⁻⁸ RIU sensitivity is interesting, but I am troubled about the claimed resolution and a missing graph that I expected.

As we were not clear enough in the previous version of the manuscript, we have changed a paragraph on page 11, as follows: *The amplifier was set close to its maximum so that the sound level (corresponding to $20\log \frac{p}{2e^{-5}}$ with p expressed in Pa [25]) was measured to be 109 dB at the sensor location, using the XL2 analyzer from NTI Audio. According to the aforementioned formula, this yields an atmospheric pressure change of only 5.6 Pa, corresponding to an air refractive index change as small as $1.48 \cdot 10^{-8}$ RIU at constant temperature (Eq. 2).*

Page 3, para 3 and 4:

I would suggest that reference 8 is not applicable to the introductory text discussing D-shaped fibers and plasmons, but rather M. Zervas and I. Giles, "Performance of surface-plasma-wave fiber-optic polarizers," Opt. Lett. 15, 513-515 (1990), is a more suitable reference.

I would also dispute that polishing weakens fibers, as there are many examples of quality polished components where fibers are embedded in glass blocks, prior to polishing, forming a robust, monolithic structure after polishing. Does the fiber, which the authors use, not have to be stripped to be coated with Au, where is the robust nature here? I think that general statements of this type in support of an apparent advantage rarely offer anything.

Agreed. This statement has been removed from the introduction and reference [8] has been changed following the suggestion.

I think that the use of the expression "privileged" when discussing reference 9, it is not used conventionally and is confusing.

Agreed. It has been changed to "preferred".

Page 4

Last paragraph is incorrect, there are fibre geometries that operate in air and other gasses, see general comments and references noted above regarding this.

Indeed. This has now been changed, as follows: *Several configurations types have been proposed to overcome this limitation. Multilayer coatings have been used on top of unclad fiber sections to provide a medium with correct refractive index for SPP excitation [24-26]. Sensing mechanisms are then based on the refractive index changes that occur in this multilayer structure when gases are adsorbed into it (the adsorption can be chemically specific depending on the material that is used), which in turn will modify the SPP. A tapered fiber optic tip sensor with angled facets such that optical modes with effective index close to 1 can be excited has also been proposed few years ago [27,28].*

Page 5

OK

Page 6

General information is OK here and of sufficient detail, although a comment regarding how accurately the polarization "axes" must be aligned to would be necessary; again I think there are other papers to reference in addition to [11]. How is the system affected by slow/sudden polarization changes? How stable is the sensor over time?

OK. The following sentences have been added as well as two new references [13,14]: Note that there is not only one polarization state that yields SPP excitation but rather a set. For instance, when linear polarization states are used, a range of 10° can be used for clean SPP excitation in the amplitude transmitted spectrum of the TFBG.

Concerning the polarization changes, it is clear that such systems are affected by changes in the polarization drift over time. It is the reason why care is taken to avoid polarization instabilities: connecting fibers are kept as short as possible and they are fixed. In our case, measurements were conducted over limited time periods (not exceeding a few seconds) so that polarization drift was not an issue.

That word "privileged" appears again.

Figure 1: Good figure.

It has been changed to "preferred".

Page 7

Nice explanation and crystal clear.

Figure 2: Good figure.

Page 8

Good explanation, good figure (Fig. 3).

Page 9

Figure 5, "caldding" spelling error. I would also suggest that the comparative figures should have the same scale on the wavelength axes.

It has been corrected.

Page 10

Figure 6, "caldding" spelling error.

It has been corrected.

In Fig 5 the authors show the activated cladding mode, for which they also perform numerical analysis as a form of "confirmation", is located a 1325nm, but in the measurements of Fig 6 they select a cladding mode close to 1316nm. It is critical to define which modes are selected and why, is a wavelength shift preferential or a loss measurement? Surely the former mitigates any intensity fluctuations in the system, which could prove very important, as the changes in amplitude (and wavelength) are small. If the intensity change is self-referenced then that should be made clear. Whatever the case an explanation would be useful.

The difference in wavelength arises from the fact that both experiments were conducted on two different gratings for which the resonances do not match exactly. From the manufacturing

process, a difference in wavelength of a few nanometers between gratings is possible. This difference can be further increased by a slight change of gold coating thickness (see ref. [34] for instance).

To better clarify this point, the paragraph in page 10 has been modified as follows: *Other measurements were performed on another grating with smaller atmospheric pressure changes to quantify the sensitivity of the SPP-matched cladding mode resonance to air pressure (and hence to the refractive index of the fiber surrounding). Figure 6 depicts a spectral region around the SPR signature, confirming that only the subset of modes that are phase matched with the SPR envelope are sensitive to the atmospheric pressure change, with different degrees of sensitivity, as already observed for refractometric changes in liquids [34]. The inset of Fig. 6 focuses on the evolution of the most sensitive cladding mode resonance, showing again a clear wavelength shift and amplitude change. Similarly to our works conducted in liquids, this mode is the most sensitive as it is located on the shoulder of the SPR envelope. Both the wavelength shift and amplitude change can be used to estimate the SPP modification induced by surrounding refractive index changes. The first one is inherently insensitive to unwanted optical power fluctuations. The second is also immune thanks to a self-referencing of the spectrum with the Bragg resonance that remains unaffected by surrounding refractive index changes, as demonstrated in Figure 5.*

Page 11

The final experiment is crucial to this work; it is the justification for this paper. However, here I am puzzled by the results that are presented. Let me explain why. Fig 6 shows the most sensitive response to pressure for which a 30kPa pressure change results in an intensity change of < 1dB, it looks closer to 0.5dB, or 1.67×10^{-5} dB/Pa, or $\sim 10^{-4}$ RIU. The raw data is plotted in Fig 7 and the error bars suggest an error far greater than the claimed measurement resolution. The source of the errors is critical but not stated, although it could possibly be linked to the ability to control the pressure and/or temperature during the experiment. Hence this should be addressed.

Now an experiment is undertaken where the pressure change is limited to 5.6Pa or an equivalent intensity change of 9×10^{-5} dB, or $\sim 1.8 \times 10^{-8}$ RIU. This may be anywhere from 100 to 1000 times smaller than the error estimation of Fig 7. Another high-resolution measurement method is used to record the low pressure data. The data in Fig 8 looks good with the two curves tracking each other, but where is the graph showing amplitude (pressure change) versus RIU? I would suggest that such a graph is paramount for the low pressure scale (that would also have the "same" gradient as Fig 7 if all is correct and calibrated) and would be most appropriate in order to make this final, and most important, result convincing. Furthermore, more information is required regarding the speaker location, acoustic focussing and positioning of the sensor relative to the acoustic wave.

The resolution of the "static" measurement and calibration shown in Fig. 7 is severely limited by the instrumentation: in order to measure a 10^{-8} RIU change, wavelength shifts of 0.003 pm would have to be measured, which is impossible with an optical spectrum analyzer. Therefore the only way to "prove" that the TFBG-SPR can achieve such high resolution is by a measurement that avoids most sources of noise, i.e. a dynamic measurement of an oscillating RIU, i.e. a sound wave.

The following sentence has been revised on page 12: *The distance between the loudspeaker (6 cm of diameter without acoustic focusing) and the sensor was 15 cm.* The fiber orientation with respect to the sound wave is the one indicated in Fig. 1.

To summarise, the uniqueness of this work relies on the use of SPR cladding modes generated by highly tilted gratings, to measure low indices and hence pressure. The work draws extensively on earlier research by the same authors to develop their sensor system. The key result is an acoustic measurement for a low pressure wave that is associated with a RIU resolution of $\sim 10^{-8}$. I believe that some more work is required to remove any possible sources of error in this very delicate and potentially tricky experiment. This concludes my review.

Reviewer #3 (Remarks to the Author):

The authors C. Caucheteur et al propose and demonstrate highly sensitive surface plasmon resonance (SPR) sensors in air by using standard single mode optical fibers inscribed with tilted Bragg gratings. Based on the tilted Bragg gratings, SPR can be excited at the interface between the optical fiber cladding and the metal coating, which is extremely sensitive to the refractive index (RI) changing at external RI of around 1.0. Attributing to the narrow transmission spectrum of tilted Bragg gratings, the sensitivity of the proposed SPR-based sensor to external gas RI can be significantly improved, which is about 10⁻⁸ RIU near the RI of 1.0. To the best of our knowledge, such sensitivity is the highest one based on SPR technique.

In the background introduction, the authors emphasize that the fiber-based SPR sensors have never used in gaseous phases (line 4, paragraph 3, page 3). However, some structures like prism-coupling have been introduced into optical fiber for the vapor phase sensors, as shown in the following references. But the reported sensitivity is relatively low due to the wide band SPR spectrum.

Ref.1 Yoon-Chang Kim, Soame Banerji, Jean-Francois Masson, Wei Peng and Karl S. Booksh, Fiber-optic surface plasmon resonance for vapor phase analyses, *Analyst*, 2005, 130, 838-843.

Ref.2 Yoon-Chang Kim, Wei Peng, Soame Banerji, and Karl S. Booksh, Tapered fiber optic surface plasmon resonance sensor for analyses of vapor and liquid phases, *OPTICS LETTERS*, 2005, 30(17):2218-2220

Agreed. The introduction has been modified as follows: *The development of optical fiber SPR platforms for gas sensing where the refractive index of the medium is far from the one of the waveguide is not trivial. Hence, unclad optical fiber and fiber grating SPR configurations reported so far operate well in aqueous solutions (i.e. in any media with refractive indices between 1.30 and ≈1.43) but are not able to provide direct (i.e. with only a single metal layer on the silica surface) SPP excitation in air or gaseous environments. Indeed, most of the configurations reported so far are unable to couple light into fiber modes that have sufficiently small effective refractive indices (corresponding to large enough incidence angles at the cladding outer boundary). Several configurations types have been proposed to overcome this limitation. Multilayer coatings have been used on top of unclad fiber sections to provide a medium with correct refractive index for SPP excitation [24-26]. Sensing mechanisms are then based on the refractive index changes that occur in this multilayer structure when gases are adsorbed into it (the adsorption can be chemically specific depending on the material that is used), which in turn will modify the SPP. A tapered fiber optic tip sensor with angled facets such that optical modes with effective index close to 1 can be excited has also been proposed few years ago [27,28].*

In this paper, we demonstrate a quite simple and robust all-fiber configuration able to excite SPP directly in air with narrowband cladding mode resonances (linewidth < 200 pm) that can be measured with a high resolution.

The explanation of two-resonator system is unclear. The authors should give which order cladding modes are excited in the SPR wavelength band. Because the effective RI of any fiber modes cannot be less than 1, the detailed explanation about the cladding mode resonance is of importance to convince the reader that the cladding mode resonances can present effective RI ranging between 0.92 and 1.18.

In more detail, the “effective index” of a mode is given by the ratio of the phase velocity in vacuum over the phase velocity of the mode in the fiber, MEASURED ALONG THE FIBER AXIS. Taking the analogy of a plane wave incident on a flat interface between a high refractive index medium and a low refractive index medium, a similar effective index can be calculated for the phase velocity along the interface. As the angle of incidence decreases from 90 degrees towards the critical angle the phase velocity along the interface increases (and the effective index decreases). Passing the critical angle, the wave becomes leaky but the phase velocity along the interface continues to increase. Indeed, as the angle of incidence reaches zero, the phase velocity along the interface reaches infinity. The same phenomenon occurs in fibers, or at the base of a Kretschmann-Raether prism, as the angle of incidence decreases (equivalent to the order of the cladding mode increasing). For a fiber in air (or vacuum), effective indices lower than 1.0 simply mean that the corresponding cladding mode is leaky. Simulations indicate that the modes with effective indices ~ 1.0 have radial orders near 167. Furthermore it is already indicated in the manuscript (before figure 4) that the effective RI of a surface plasmon wave at the interface between gold and air is equal to 1.007 at these wavelengths.

The temperature as well as the polarization perturbation to the accuracy of sensors' sensitivity should be clearly illustrated. As given in Eq. (2) in page 10, the RI of air is inversely proportional to the ambient temperature (T). Does the crosstalk from temperature influence the SPR cladding mode resonance? In addition, the proposed titled FBG is polarization-sensitive. Does the polarization perturbation affect the SPR cladding mode resonance and thus the sensors' sensitivity?

Concerning temperature changes, they will clearly affect the response, as they are responsible for changes in the surrounding refractive index medium. They were not considered in this pioneer work for which dynamic measurements were conducted in a thermally-stabilized clean room environment (class 10,000) over limited periods of time (not exceeding a few seconds). The influence of the temperature on the fiber itself (because of the thermo-optic coefficient of glass and the thermal expansion of the grating period) is removed from the measurements by referencing all wavelengths to the resonance of the core mode reflection (which senses temperature and strain but not SRI). This information has been added on page 9.

Concerning the polarization changes, it is clear that such systems are affected by changes in the polarization drift over time. It is the reason why care is taken to avoid polarization instabilities: connecting fibers are kept as short as possible while they are fixed. For the reasons outlined above, polarization drift was not an issue in our measurements.

Further discussion about the limitation of the used equipment on the accuracy of sensors' sensitivity is required. The wavelength shift of the proposed SPR sensor is less than 0.01 nm as shown in Fig. 7, however, the highest resolution of optical spectrum analyzer (AQ6370) is 0.01 nm. Do the wavelength resolution limitation of OSA and the sensitivity of power-meter influence the accuracy of the obtained sensors' sensitivity?

Indeed. As discussed above in response to another comment from Reviewer 2, it is the reason why high-resolution measurements have been performed with a narrowband laser source and a photodiode. Optical spectrum analyser measurements are only used for visualization purposes.

The typos need to be revised:

"1.48 10⁻⁸" in line 7, paragraph 1, page 11

"calding" in the annotation of Fig. 5 and Fig. 6s explained in [19].

They have been corrected.

Reviewers' comments:

Reviewer #1 (Remarks to the Author):

The authors have responded clearly to questions posed. The answers are clear and enlightening, leaving no room for doubts. All the additional text was properly included, enhancing the quality of the paper. Therefore, the paper should be approved for publication.

Reviewer #2 (Remarks to the Author):

This review (reviewer 2) is based on the changes made by the authors with respect to the opinions of three reviewers. I will judge their response to all points raised, as they are very similar between the reviewers, and note that the authors have addressed the majority of the points raised by the reviewers, principally through text changes and new and relevant references. There is one point that has not been addressed that I had raised. My concern was regarding figure 8; it appears that the authors consider that this figure is self explanatory. However, there was no graph associated with this figure, which would map the anticipated behavior according to equation 2. Hence we have figure 8 that is one point on a graph that is related to the index described by equation 2. I still believe that the critical point of this paper relates to this missing graph, and strongly believe that the authors need to link index changes to changes in external pressure for their second experiment. This would then be conclusive "proof" of the claimed resolution of 10⁻⁸ RIU.

Reviewer #3 (Remarks to the Author):

The revised paper has been significantly improved regarding our concern about the state of the art, the working mechanism, and the accuracy of the sensitivity of the sensor. One question is raised about the sensitivity of the sensor influenced by temperature fluctuation. It is addressed that the influence of the temperature on the fiber itself is removed from the measurements by referencing all wavelengths to the resonance of core mode reflection (line 238-241 in page 9). However, cladding mode should have a higher sensitivity than the core mode to the change of temperature and strain. Therefore, the detailed explanation about the sensitivity of cladding mode with the change of temperature and strain is required.

Moreover, in order to improve the quality of the paper rather than just reply to the reviewers' comment, it would be better to add all the information addressed in the response letter to the submitted manuscript. For instance, which order cladding modes is excited in the SPR wavelength band? What is the theoretical model?

Reviewer #1 (Remarks to the Author):

The authors have responded clearly to questions posed. The answers are clear and enlightening, leaving no room for doubts. All the additional text was properly included, enhancing the quality of the paper. Therefore, the paper should be approved for publication.

We warmly thank the reviewer for this positive feedback.

Reviewer #2 (Remarks to the Author):

This review (reviewer 2) is based on the changes made by the authors with respect to the opinions of three reviewers. I will judge their response to all points raised, as they are very similar between the reviewers, and note that the authors have addressed the majority of the points raised by the reviewers, principally through text changes and new and relevant references. There is one point that has not been addressed that I had raised. My concern was regarding figure 8; it appears that the authors consider that this figure is self explanatory. However, there was no graph associated with this figure, which would map the anticipated behavior according to equation 2. Hence we have figure 8 that is one point on a graph that is related to the index described by equation 2. I still believe that the critical point of this paper relates to this missing graph, and strongly believe that the authors need to link index changes to changes in external pressure for their second experiment. This would then be conclusive "proof" of the claimed resolution of 10^{-8} RIU.

We agree with the reviewer that this point remained the most critical one of the revised version of the manuscript. We have therefore conducted a new set of experiments for which the sound level provided by our system was tuned, allowing us to better determine the limit of detection (LOD) of the whole sensor configuration (gold-coated highly-tilted FBG + dynamic interrogator). The results presented in Figure 8 were obtained for a 2 kHz sine excitation of 109 dB, very close to the maximum value that can be delivered by the system. In the new experiments that were conducted, the sound level was decreased by steps of ~ 1 dB (measured by the NTI Audio analyzer) starting from the maximum level and the sensor response was recorded accordingly. It turns out that the following results were obtained:

Sound level (dB)	Pressure change (Pa)	Refractive index change (RIU)	Mean peak-to-peak amplitude variation
105.0 \pm 0.1	3.56 \pm 0.04	0.94 \pm 0.02 10^{-8}	--
106.1 \pm 0.1	4.04 \pm 0.05	1.07 \pm 0.02 10^{-8}	0.0036 \pm 0.0010
107.1 \pm 0.1	4.53 \pm 0.05	1.20 \pm 0.02 10^{-8}	0.0078 \pm 0.0012
108.0 \pm 0.1	5.02 \pm 0.06	1.33 \pm 0.03 10^{-8}	0.0119 \pm 0.0018
109.2 \pm 0.1	5.77 \pm 0.06	1.53 \pm 0.03 10^{-8}	0.0208 \pm 0.0024

Hence, for a sound level below 105 dB (this value corresponds to an atmospheric pressure change of 3.56 Pa and an air refractive index change of $0.94 \cdot 10^{-8}$ RIU), the sensor trace recorded by the oscilloscope remained flat. A growing sine function at 2 kHz frequency was then recorded for sound levels above 106 dB. The measurements were conducted until the saturation (limit of excitation - LOE) of our system was reached, corresponding to a sound level measured at 109.2 dB. For each investigated sound level, 5 measurements were recorded and the results presented in the figure below are the mean and standard deviation of these 5 measurements. From these results, we can confirm that the LOD of our sensor configuration is very close to 10^{-8} RIU.

These explanations and the picture have been added in the manuscript at the end of the experimental section.

Reviewer #3 (Remarks to the Author):

The revised paper has been significantly improved regarding our concern about the state of the art, the working mechanism, and the accuracy of the sensitivity of the sensor. One question is raised about the sensitivity of the sensor influenced by temperature fluctuation. It is addressed that the influence of the temperature on the fiber itself is removed from the measurements by referencing all wavelengths to the resonance of core mode reflection (line 238-241 in page 9). However, cladding mode should have a higher sensitivity than the core mode to the change of temperature and strain. Therefore, the detailed explanation about the sensitivity of cladding mode with the change of temperature and strain is required.

We thank the reviewer for his positive remark about the quality progress of our manuscript. We have studied the temperature and axial strain sensitivities of the cladding modes several years ago [A]. It turns out that the temperature sensitivity stays almost constant whatever the order of the cladding mode resonance. For the axial strain sensitivity, it decreases with the mode order.

These observations on weakly tilted FBGs are further confirmed by other results recently obtained with eccentric FBGs (refraction index modulation created close to the core-cladding interface) produced by a femtosecond pulses laser. A manuscript about these results is currently under review.

As axial strain effects were not considered in this work, we have added the following information in page 11:

Temperature changes were not considered in this pioneer work for which dynamic measurements were conducted in thermally-stabilized clean room environment over limited periods of time not exceeding a few seconds. In such small time periods, the ambient temperature can be considered as constant. Note that the influence of the temperature on the fiber itself (because of the thermo-optic coefficient of glass and the thermal expansion of the grating period) is removed from the measurements by referencing all wavelengths to the resonance of the core mode reflection (that senses temperature and strain but not SRI) [11].

[A] C. Chen et al, "The sensitivity characteristics of tilted fibre Bragg grating sensors with different cladding thicknesses," Meas. Sci. Technol. **18**, 3117-3122 (2007).

Moreover, in order to improve the quality of the paper rather than just reply to the reviewers' comment, it would be better to add all the information addressed in the response letter to the submitted manuscript. For instance, which order cladding modes is excited in the SPR wavelength band? What is the theoretical model?

Agreed. The following paragraph has been added in page 7:

In more detail, the effective refractive index of a mode is given by the ratio of the phase velocity in vacuum over the phase velocity of the mode in the fiber, measured along the fiber axis. Taking the analogy of a plane wave incident on a flat interface between a high refractive index medium and a low refractive index medium, a similar effective index can be calculated for the phase velocity along the interface. As the angle of incidence decreases from 90 degrees towards the critical angle the phase velocity along the interface increases (and the

effective index decreases). Passing the critical angle, the wave becomes leaky but the phase velocity along the interface continues to increase. The same phenomenon occurs in fibers, or at the base of a Kretschmann-Raether prism, as the angle of incidence decreases (equivalent to the order of the cladding mode increasing). For a fiber in air (or vacuum), effective indices lower than 1.0 mean that the corresponding cladding mode is leaky.

At the bottom of page 8, it has been specifically written:

Grating spectra can be studied and predicted using the coupled mode theory [34]. In our case, numerical simulations conducted using the finite difference mode solver FimmWave (from Photon Design Inc.) were carried out to confirm the experimental evolutions. (...) Simulations indicate that the modes with effective indices ~ 1.0 have radial orders near 167.